



# Extreme waves and climatic patterns of variability in the Eastern North Atlantic and Mediterranean basins

Verónica Morales-Márquez[1], Alejandro Orfila[1], Gonzalo Simarro[2], and Marta Marcos[1]

[1]IMEDEA (UIB-CSIC), Esporles, Balearic Islands, Spain
[2]ICM, 08003 Barcelona, Catalonia, Spain

**Correspondence:** Verónica Morales-Márquez (vmorales@imedea.uib-csic.es)

**Abstract.** The spatial and temporal variability of extreme wave climate in the North Atlantic Ocean and the Mediterranean Sea is assessed using a 31-year wave model hindcast. Seasonality accounts for $50\%$ of the extreme wave height variability in North Atlantic Ocean and up to $70\%$ in some areas of the Mediterranean Sea. Once seasonality is filtered out, the North Atlantic Oscillation and the Scandinavian Index are the dominant large-scale atmospheric patterns that control the interannual

variability of extreme waves during winters in the North Atlantic Ocean; and to a lesser extent, the East Atlantic Oscillation also modulates extreme waves in the central part of the basin. In the Mediterranean Sea, the dominant modes are the East Atlantic and East Atlantic/Western Russia modes which act strongly during their negative phases.

## 1   Introduction

The accurate assessment of extreme wind-wave conditions is essential for human activities e.g., maritime traffic and wave energy generation and is a major source of coastal hazards. Besides, extreme waves influence the upper ocean by enhancing vertical mixing through the Stokes layer. Extreme waves reaching port areas, determine the design and operation of coastal and offshore infrastructures, they are also responsible for coastal flooding at intra-annual scales. Waves are the ocean surface response to the wind stress acting over it and therefore there is a direct connection between surface atmospheric circulation and

waves (Lin et al., 2019).

The study of extreme waves at different temporal scales has been extensively addressed in several works (Wang and Swail, 2001, 2002; Caires et al., 2006; Méndez et al., 2006; Menéndez et al., 2008, 2009; Izaguirre et al., 2010, 2012; Young et al., 2012; Weiss et al., 2014; Sartini et al., 2017). Most of these studies focused on the spatio-temporal distribution of extreme waves rather than on the atmospheric conditions producing them.

Over the North Atlantic Ocean and the Mediterranean Sea, atmospheric circulation is driven by the temperature gradient between the North Pole and the Equator that organizes three-cell system associated with the Equatorial-Low, the subtropical Azores-High, the Iceland-Low and the North Pole-High pressure centres (Martínez-Asensio et al., 2016). Atmospheric cir-





culation can be characterized by specific modes of variability with defined characteristics that may also have effects over a region or remote areas through atmospheric teleconnections (Wallace and Gutzler, 1981). The main patterns of atmospheric

variability on the North Atlantic and Europe are the North Atlantic Oscillation (NAO), the East Atlantic pattern (EA), the Scandinavian pattern (SCAND) and the East Atlantic/Western Russia (EA/WR) patterns (Barnston and Livezey, 1987). NAO is the leading mode of variability in the North Atlantic and is often defined as the sea level pressure difference between the Iceland Low and the Azores High (Hurrell et al., 2003). NAO controls the strength and direction of westerly winds reaching the European coasts, as well as the location of the storm tracks across the North Atlantic (Marshall et al., 2001). The EA is

the second predominant mode of low frequency variability in the North Atlantic area. It consists of a north-south dipole of anomaly over the North Atlantic, with a strong multidecadal variability. The SCAND pattern consists of a primary circulation center over Scandinavia, with weaker centers of opposite sign over western Europe. The EA/WR pattern consists of four main anomaly centers; its positive phase is associated with positive wave height anomalies located over Europe and negative wave height anomalies over the central North Atlantic (https://www.cpc.ncep.noaa.gov/data/teledoc/telecontents.shtml).

There have been a number of studies that have tried to unravel the relation between wave climate and large scale atmospheric patterns. Woolf et al. (2002) found a strong connection between interannual wave climate variability in North Atlantic Ocean and NAO, and in a lesser degree with EA index. Castelle et al. (2018) examined the relation between winter-mean wave height, detailing a high correlation with the NAO index and with the Western Europe Pressure Anomaly (WEPA) index; this is a new definition of a climatic pattern which based on the sea level pressure gradient between the stations Valentia (Ireland) and Santa

Cruz de Tenerife (Canary Islands). Izaguirre et al. (2010) detected a relation between the NAO and EA indices with the extreme wave climate in the North-East Atlantic Ocean. Izaguirre et al. (2012) evaluated the synoptic atmospheric patterns associated to the extreme Significant Wave Height (SWH) finding a higher interannual variability of the extreme SWH in the northern part of the Atlantic Ocean. In the Mediterranean Sea, clear relations between extreme waves and the negative phases of EA and the EA/WR indices have been also reported (Izaguirre et al., 2010).

In this paper, we extend earlier studies, analyzing the short and long term variability of extreme waves in the North Atlantic Ocean and in the Mediterranean Sea, not only for diagnostic purposes but also to be able to provide statistical prognostics of extremes waves associated to the most important climatic indices with some anticipation. The paper is structured as follows. In Section 2, the data used and the description of the extreme waves are presented. In Section 3, we present the spatial and temporal distribution of the extreme waves as well as the relation between the four patterns of climatic variability and the

spatial distribution of extreme waves during winter. Finally, Section 4 concludes the work.

## 2  Data and methods

### 2.1  Waves and atmospheric data

Wave data is obtained from a high resolution global hindcast from the National Center for Environmental Prediction (NCEP) with a temporal sampling of 1 hour and different spatial resolutions. This dataset (i.e.,*WAVEWATCH III 30-year Hindcast Phase*

*2*, (Chawla et al., 2012)) has been generated by forcing the "state-of-the-art" wave model WAVEWATCH III (Tolman, 2009)





with 10-m height high-resolution wind fields from the NCEP Climate Forecast System Reanalysis and Reforecast (CFSRR) a 30-year homogeneous data set of hourly $1/2°$ spatial resolution winds.

The wave model consists of global and regional nested grids, developed by the presence of currents and bathymetry (Amante and Eakins, 2009), taking into account the conservation of action density (Janssen, 2008). In addition, the dissipation and physical terms parameterization formulated inArdhuin et al. (2010) is used in this work.

The hindcast has been validated using the National Data Buoy Center (NDBC-NOAA) as well as with altimeter database provided by the Institut Français de Recherche pour l'Exploitation de la Mer (IFREMER) database (Chawla et al., 2011, 2012, 2013).

The simulation spans a time period of 31 years from 1979 to 2009 with hourly outputs, although since we assume that wave climate is constant for 3 hours, we only use 1 data for this period; and with a spatial resolution varying according to the study area. In all the grids, the full resolution ETOPO1 bathymetry is used in regular spherical grids. The North Atlantic domain spans from $20°$N to $70°$N in latitude and $60°$W to $10°$E in longitude at $0.5°$ resolution (Fig. 1a). The Mediterranean Sea covers from $30°$N to $48°$N in latitude and $7°$W to $43°$E in longitude with a spatial resolution of $0.167°$ (see Fig. 1b). Sea level pressure and wind velocity at 10 meters height are provided by the NCEP-CFSR forcing with a resolution of $0.5°$ for the same period (Saha, 2009).

Leading climatic modes of variability, namely NAO, EA, EA/WR and SCAND (see introduction Section for a description of these modes) have been downloaded from the NOAA Climate Prediction Centre. Indices are constructed through a rotated principal component analysis of the monthly mean standardized $500$-mb height anomalies in the Northern Hemisphere ensuring the independence between modes, at a monthly scale, due to orthogonality (Barnston and Livezey, 1987).

## 2.2 Extreme wave climate

Extreme wave climate is here defined in terms of the monthly $99^{\text{th}}$ percentile of SWH (hereinafter $\text{SWH}_{99}$). Over the North Atlantic, maximum values of $\text{SWH}_{99}$ during the whole period of time analyzed (1979-2009), are observed at mid to high latitudes with values reaching 13 m (see Fig. 2a). These situation is very similar to the obtained in Vinoth and Young (2011), where there are higher values of SWH in the northern part of the study area. These maximum values befall predominantly during winter season (DJFM), with an $81.2\%$ of occurrence (Fig. 2c). Over the Mediterranean Sea, maximum values of $\text{SWH}_{99}$ are at most 8 m (Fig. 2b) with a $91.06\%$ of occurrence during winter season (DJFM) (Fig. 2d).

Seasonality is assessed by fitting a cosine function to the monthly $\text{SWH}_{99}$ series through a least squares adjustment (Menéndez et al., 2009),

$$f(t) = \sum_{i=1}^{2} A_i \cos\left(\frac{2\pi}{T_i}(t - \phi_i)\right): \tag{1}$$

where, $i = 1, 2$ are the annual and semiannual cycle, $A_i$ the amplitude, $\phi_i$ the phase, $T_{1,2} = 365.25$ and $182.63$ and $t$ time in days. The monthly $\text{SWH}_{99}$, for the location with the largest variance reduction in the North Atlantic (point 1, Fig. 3a), is shown in black in the top panel of Fig. 3 (the same for the Mediterranean Sea, point 2 in Fig. 3b in the bottom panel), while the time series after removing seasonality by fitting Eq. (1) are shown in blue. Seasonality in the North Atlantic accounts on average for


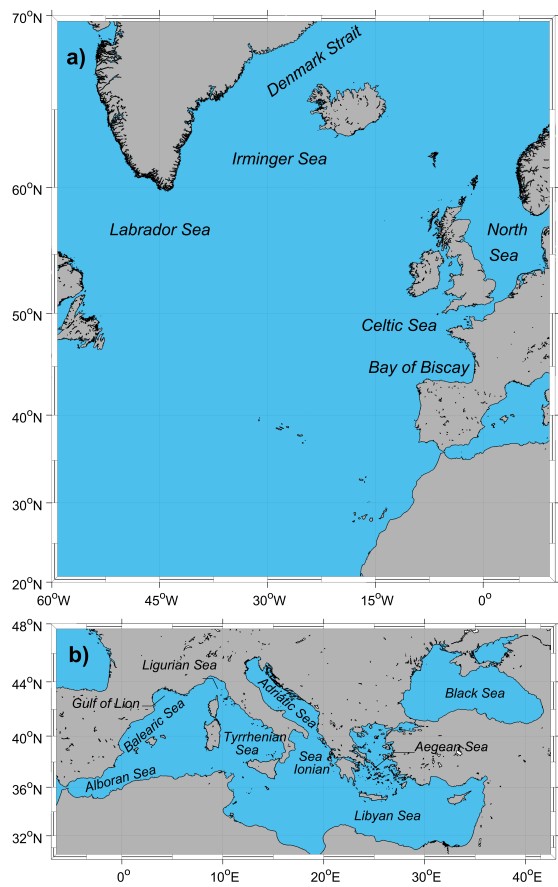

**Figure 1.** Location of study zones. a) Eastern North Atlantic Ocean. b) Mediterranean Sea.

a 50% of the variance of the signal (Fig. 3a). It means that the half of the extreme waves signal is explained with the annual and
semiannual cycle. In the Mediterranean Sea, there are two different areas in terms of seasonality; in the central basin (where
the waves is developed) it explains up to a 70% of extreme waves while in the Gulf of Genova-Ligurian basin and Alboran Sea
seasonality explains less than 10% of the signal (Fig. 3b). These areas are very active in terms of ciclogenetic activity (Trigo
et al., 2002), where commonly the waves is generated, so the seasonal signal is relatively less important here.

To analyze the long term trend of $SWH_{99}$ during winter months (DJFM) where almost most of extreme waves occur, the
temporal series is adjusted through a first order polynomial at the 90% confidence level in each spatial point (see Fig. 4).
Locations where the trend is not significant are represented with a dot. The 31-year trend in the North Atlantic (Fig. 4a)
displays an area with significant positive values (up to 2.5 cm/year) from Portugal coasts to Canada. The rest of the basin
presents a negative value of tendency, with maximum values around 3.5 cm/year in Bay of Biscay, Labrador Sea and between
United Kingdom and Iceland. This aligns with the obtained results in Gallagher et al. (2016), where the future projections of

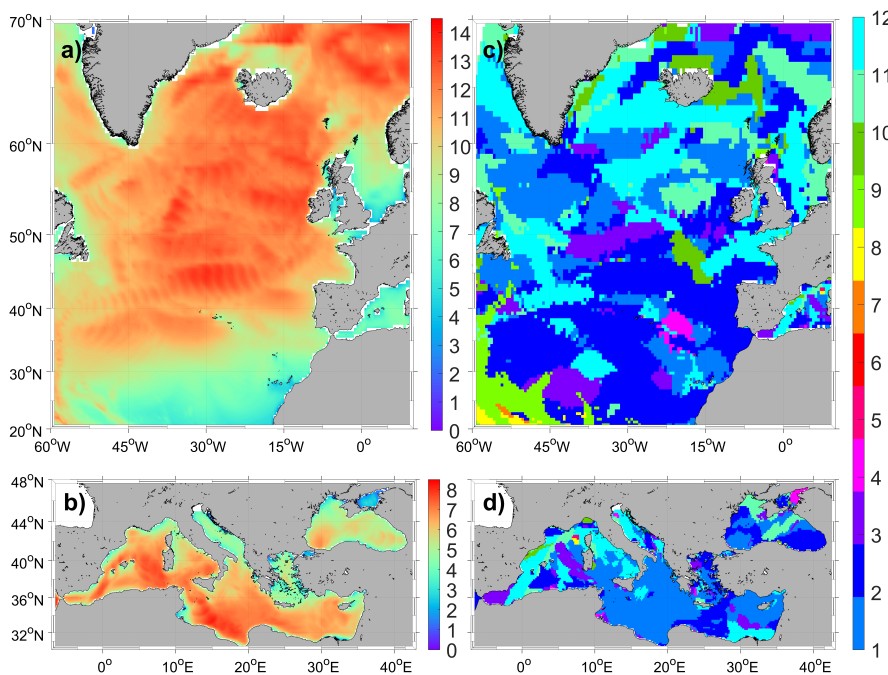

**Figure 2.** Maximum value of monthly $99^{th}$ percentile SWH in m for a) North Atlantic Ocean and b) Mediterranean Sea. And the month of the year [from January (1) to December (12)] when there is the maximum value of $99^{th}$ percentile SWH for c) North Atlantic Ocean and d) Mediterranean Sea.

mean surface wind show an average decrease over the North Atlantic Ocean for winter season. In the Mediterranean Sea, the values of $SWH_{99}$ tendency during winter months (DJFM) are substantially smaller (see Fig. 4b). Only the center part, northern of Cyprus (with negative values up to $2.4$ cm/year and $1$ cm/year, respectively) and the Aegean Sea (with positive values of trend around $1$ cm/year) present a trend statistically significant.

These results differ from the trend which is calculated in Young et al. (2011), since in that study they considered all the
months of the year in order to calculate the monthly $SWH_{99}$'s trend, and in this work, we analyze the tendency of the monthly $SWH_{99}$ only during winter season (DJFM). We have verified that there is a positive trend of $SWH_{99}$ during the summer season (it is not shown in this study), however the values in this season are considerably lower than those found during winters.

In this paper the winter extreme wave climate is studied in order to remove the seasonality, because maximum values of $SWH_{99}$ take place during the winter season.

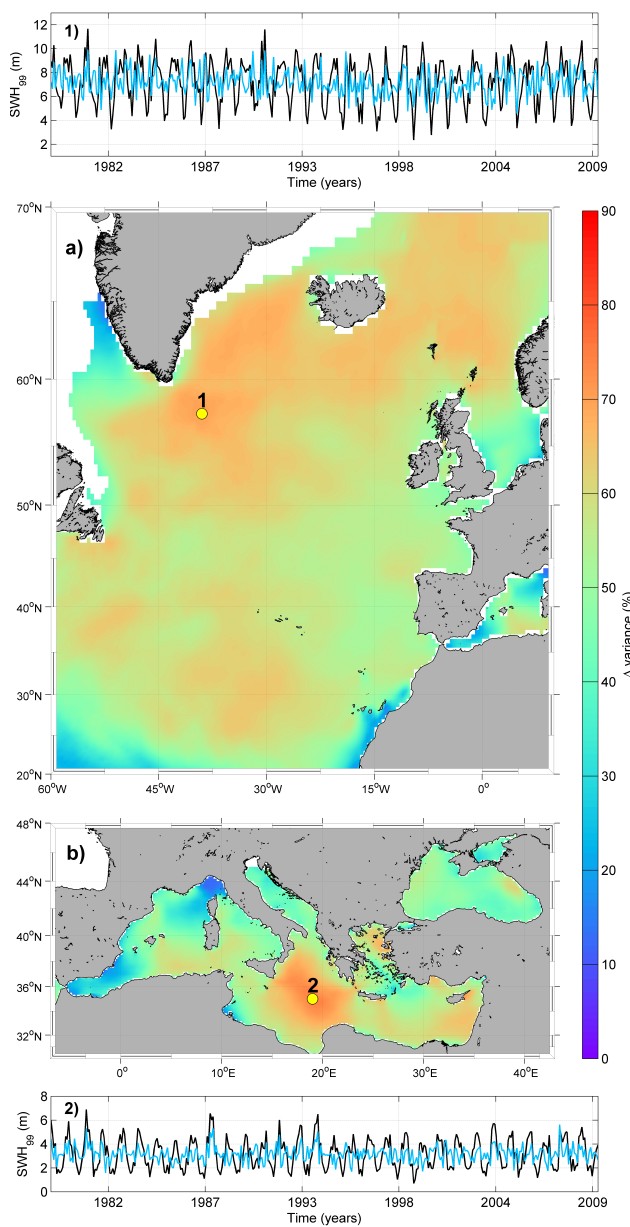

**Figure 3.** Variance reduction in percentage if the seasonality is removed of the monthly $99^{th}$ percentile SWH series for a) North Atlantic Ocean and b) Mediterranean Sea. Panels 1 and 2: monthly $99^{th}$ percentile SWH series (black line) and monthly $99^{th}$ percentile SWH series without seasonality (blue line) for a point of North Atlantic Ocean and Mediterranean Sea, respectively.



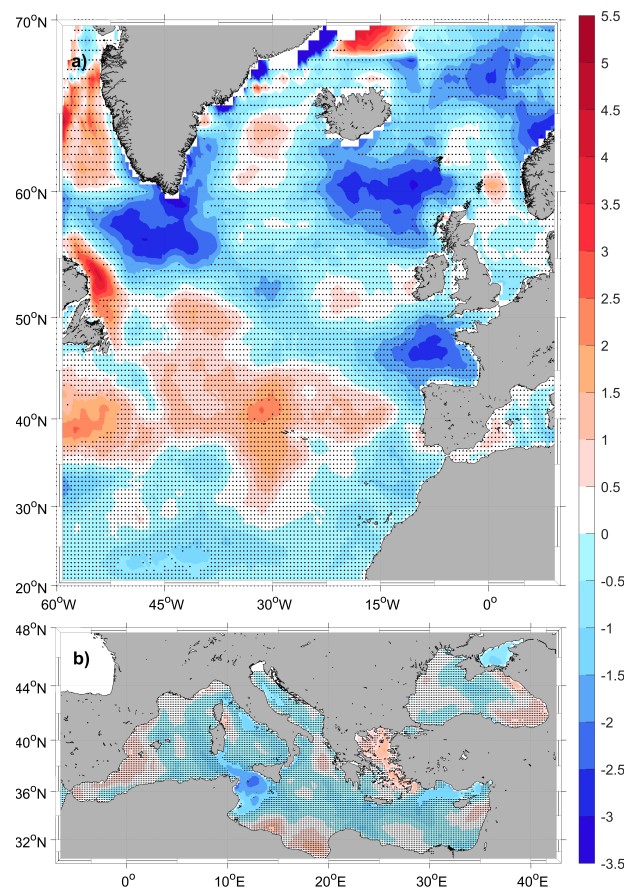

**Figure 4.** Trend of the monthly $99^{th}$ percentile SWH during winters (DJFM) in cm/year. No significant values at the $90\%$ confidence interval are dotted.

## 3 Spatio-temporal patterns of extreme waves

The spatio-temporal variability of the $SWH_{99}$ is assessed computing EOFs of the winter (DJFM) fields. Prior to the computation of the EOFs the spatial mean winter $SWH_{99}$ is removed and the analyses have been performed onto anomalies with respect to the mean values (Ponce de León et al., 2016). Mean fields are mapped in Fig. 5a and 6a.

The first three EOFs for the North Atlantic are shown in Fig. 5 (b, c and d) and their principal components (PCs) together with their explained variance in Fig. 5 (1, 2 and 3). The first EOF, which explains a $28.5\%$ of the winter $SWH_{99}$, presents a periodicity in its PC around $5$ years (Fig. 5-1). This first mode shows a spatial dipole with opposite values in the north and south of the basin. The second mode which explains a $15.5\%$ of the winter variability shows an area in the central basin separating two zones at the north and south with different sign (Fig. 5c). Values for the central part are three times larger than the ones


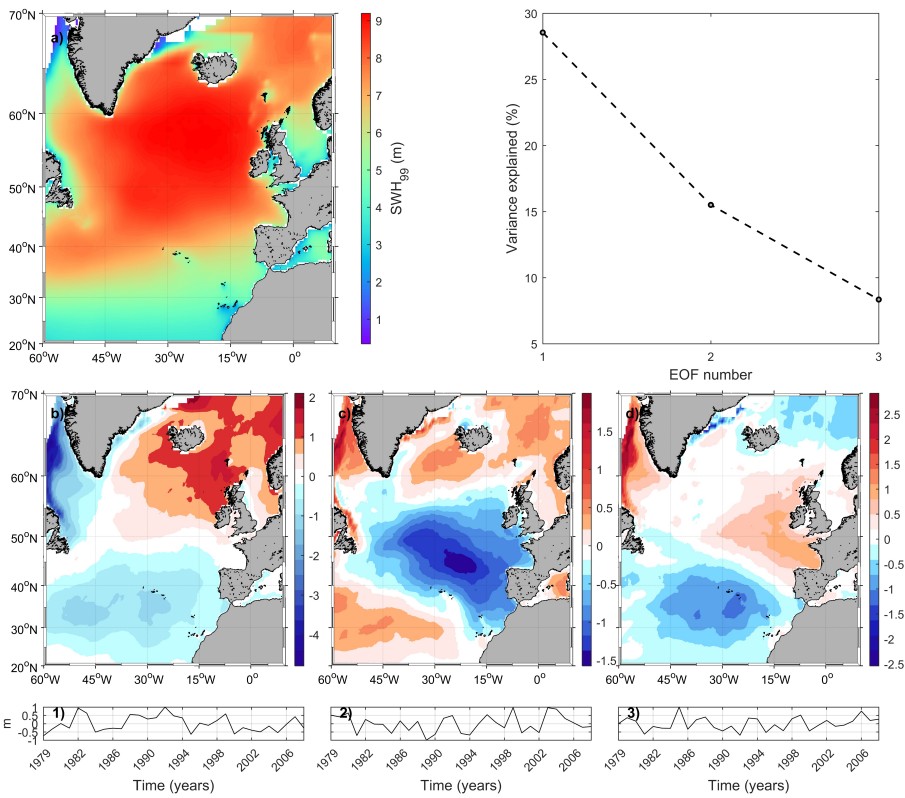

**Figure 5.** a) Mean field of winter SWH $99^{th}$ percentile over the North Atlantic. EOF analysis of $SWH_{99}$ anomalies, showing: the explained variance of the three first EOFs; b-d) spatial patterns of the EOFs 1-3; 1-3) principal components of EOFs above.

obtained for the northern and southern sides, indicating that the contribution of this EOF is to increase/decrease the winter
extremes in the central Atlantic when its PC is negative/positive. The third EOF which explains a $8.3\%$ of the winter variability displays also three different zones with a central area shifted to the East-West direction and extending from Bay of Biscay to the Celtic Sea and at the north and at the south zones displaying opposite sign during winter (Fig. 5d).

For the Mediterranean Sea, the first three EOF modes for winter are shown in Fig. 6 (b, c and d) and their PCs in Fig. 6 (1, 2 and 3). The first EOF explaining a $38.0\%$ of the total variance, represents a spatially coherent increase/decrease of $SWH_{99}$
over the entire basin. The second EOF explaining the $15.1\%$ of the variance shows differences between the eastern and western basins. The contribution of this mode is to increase/decrease $SWH_{99}$ in the western Mediterranean and simultaneously a decrease/increase in the eastern Mediterranean according to the amplitude of the PC. Finally, the third EOF explaining a $7.3\%$ of the variance displays two zones, the Tyrrhenian Sea and the southern part of the Gulf of Lion with the same sign and the rest of the basin with opposite behavior.





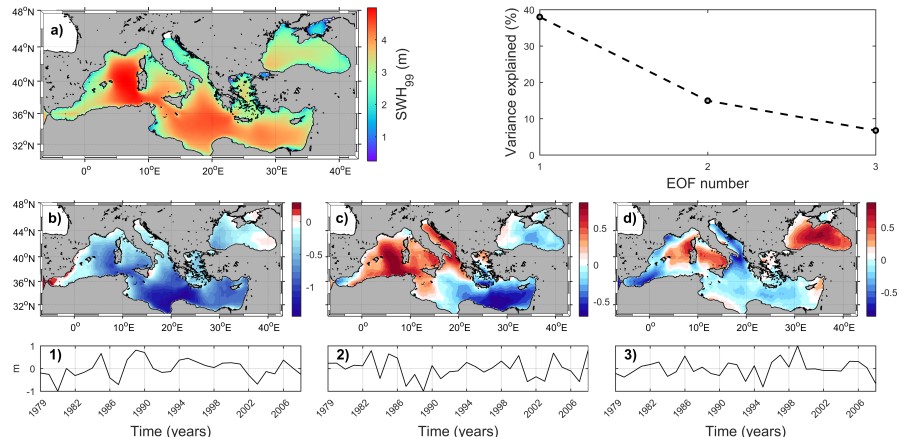

**Figure 6.** a) Mean field of winter SWH $99^{th}$ percentile over the Mediterranean Sea. EOF analysis of $SWH_{99}$ anomalies, showing: the explained variance of the three first EOFs; b-d) spatial patterns of the EOFs 1-3; 1-3) principal components of EOFs above.

The relationship between extreme waves and the climatic modes of variability in the North Atlantic and Mediterranean Seas is explored and quantified as follows. Winter averages of climate indices are first correlated with the corresponding PCs described above for each basin. The significance level is set at 90% with a $t$-value adjusted as,

$$t = |\mathrm{c}| \sqrt{\frac{N-2}{1-\mathrm{c}^2}}, \qquad (2)$$

where c the correlation coefficient and $N$ the length of the time series. If $t$ is equal or higher than the $t$-value of a Student's t-distribution of $N-2$ degrees of freedom then the correlation is assumed to be statistically significant at the predefined 90% confidence level.

### 3.1 Correlations between winter extreme waves and climatic modes of variability

The correlation between the four climate indices and the first three $SWH_{99}$ PCs are shown in Table 1 where (and hereinafter) bold indicates statistically significant correlations at the 90% confidence level. The major correlation in the North Atlantic is obtained with the NAO and the SCAND through the first PC1 (correlation of 82.6% and −63.3%, respectively). This is not surprising, as winter NAO and SCAND indices are correlated themselves (note that, although monthly indices are orthogonal, this does not necessarily hold for seasonal or yearly averages). NAO teleconnection not only dominates the extreme values of SWH during winter season; but also the mean SWH, wave period and peak wave direction magnitudes for wintertime in this region Gallagher et al. (2014). To a lesser extent, EA is correlated with the second PC (explaining around 16% of the winter variability) being the rest of correlations marginal. These results are in accordance with those obtained by Izaguirre et al. (2010) and Gleeson et al. (2019) where they show that extreme waves in the North Atlantic are related to the positive phase of NAO and with the negative of EA and SCAND. For the Mediterranean Sea, both the NAO and the EA are correlated




**Table 1.** Correlation between main climate indices and amplitudes of the three first modes of the averaged monthly $SWH_{99}$ series of the North Atlantic Ocean and Mediterranean Sea for winters.

|  | North Atlantic Ocean | | | Mediterranean Sea | | |
|---|---|---|---|---|---|---|
|  | PC1 | PC2 | PC3 | PC1 | PC2 | PC3 |
| NAO | **0.826** | $-0.138$ | **0.323** | **0.242** | $-0.193$ | **0.297** |
| EA | $-0.120$ | **-0.459** | $-0.042$ | **0.298** | **-0.361** | 0.026 |
| EA/WR | 0.057 | 0.171 | $-0.127$ | 0.093 | **-0.342** | **0.366** |
| SCAND | **-0.633** | $-0.126$ | $-0.165$ | $-0.091$ | **0.347** | **-0.390** |

with the winter $SWH_{99}$ through the first PC and some of the variability correlated by the negative phases of EA and EA/WR through the second PC. However, the values of the correlations in the Mediterranean Sea are significantly lower than those in
the North Atlantic, because the main climatic patterns consist in some strong poles located at Atlantic Ocean which drive zonal flows toward Europe. These climatic situations generate a weak circulation into the Mediterranean Sea, which is not as related to the higher values of waves. In addition, wave climate depends on wind regimes and on land-sea distribution. In other words, waves need fetch to develop and at the Mediterranean Sea, the available distance is more restricted Lionello and Sanna (2005).

**Atlantic ocean**

Correlation maps for winter $SWH_{99}$ in the North Atlantic and the four climate indices are displayed in Fig. 7. Some of these spatial correlations present similarities with the EOFs patterns shown in Fig. 5. In particular, the correlation map between NAO and $SWH_{99}$ (Fig. 7a) mimics the first $SWH_{99}$-EOF for winter (Fig. 5b) with correlation values consistent with the one obtained using PC1 (see Table 1). The correlation between EA and $SWH_{99}$ (Fig. 7b) shows large similarities with the second $SWH_{99}$-EOF for winter extreme waves (see Fig. 5c), but with the opposite sign. Finally, the correlation between SCAND and $SWH_{99}$
(Fig. 7d) shows similarities with the first $SWH_{99}$-EOF (Fig. 5b). During winter, the northern part of North Atlantic Ocean has a positive correlation with the NAO and negative at the south with maximum values close to $0.8$ (Fig. 7a). Correlation map for the EA index shows positive correlations with maximum values close to $0.75$ in the central part of the basin and negative at the north and south with maximum values of $0.47$ (Fig. 7b). Correlation map for the EA/WR displays an area of negative correlation extending from the Bay of Biscay to Greenland and also near the west coast of Africa and at the northern and central
basin appear two zones with positive correlation, with maximum values around $0.55$ (Fig. 7c). Finally, correlation map between SCAND and $SWH_{99}$ shows negative correlations in the central Atlantic with maximum correlations of $0.74$ and positive in the central Atlantic with maximum value of $0.60$ (Fig. 7d).

**Mediterranean Sea**

Correlation maps between winter $SWH_{99}$ in the Mediterranean Sea and the four climatic indices are shown in Fig. 8. Contrary
to what is found in the North Atlantic, maps of correlations between extreme waves and climate modes are not clearly linked





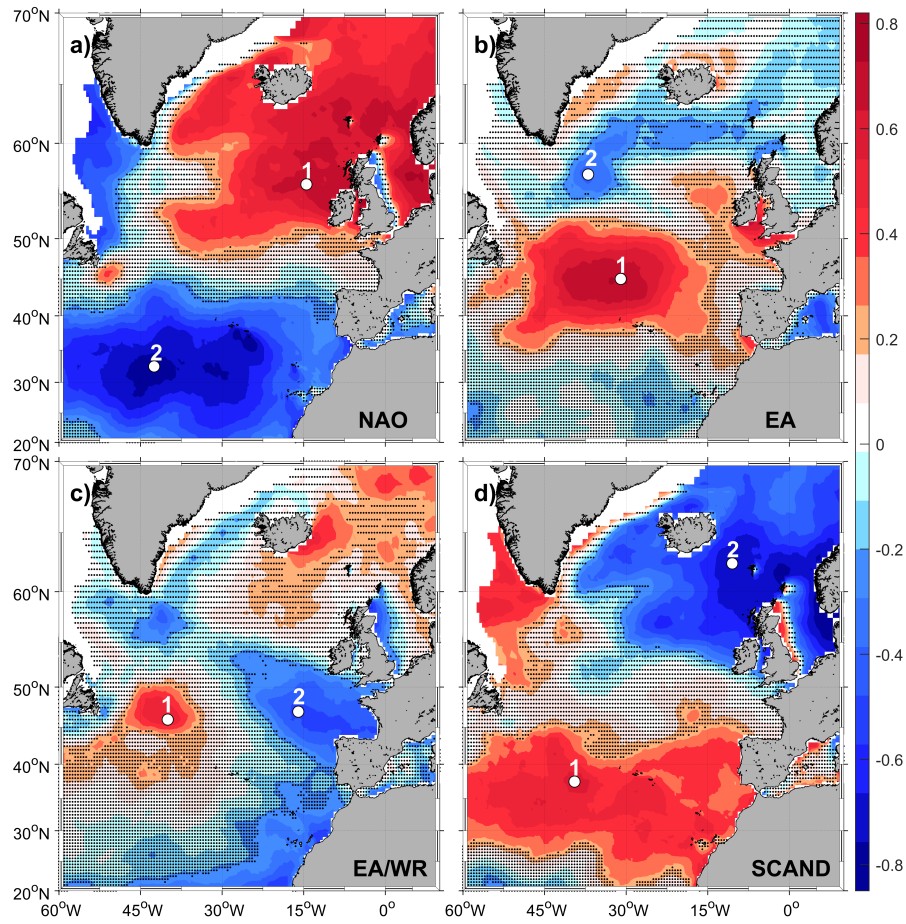

**Figure 7.** Pearson correlation coefficient of winter mean $99^{th}$ percentile SWH North Atlantic series and a) NAO, b) EA, c) EA-WR and d) SCAND winter mean indices. No significant values at the $90\%$ confidence interval are dotted. The white points show 1) the maximum positive and 2) the maximum negative value of correlation coefficient.

with the EOFs patterns of the wave field. The NAO index presents negative correlation in the whole Mediterranean basin (Fig. 8a). At the eastern side of the domain, the Adriatic and the Aegean Sea, present maximum correlations with values around $0.50$. Correlation is positive only in the Ligurian Sea, with a value around $0.30$. The EA index displays also negative correlation in the whole domain (Fig. 8b) with larger values over the west with correlation around $0.60$. The correlation map between EA/WR and SWH$_{99}$ shows negative correlations in the Tyrrhenian Sea, the Adriatic and the Ionian Seas with maximum correlation of $0.40$ (Fig. 8c). Positive correlation is obtained in the Aegean Sea with maximum values around $0.60$. Finally, the correlation map between SCAND and SWH$_{99}$ shows large positive correlations in the Gulf of Lions, at the southern and central Mediterranean Sea and in the Adriatic Sea with values of around $0.50$.





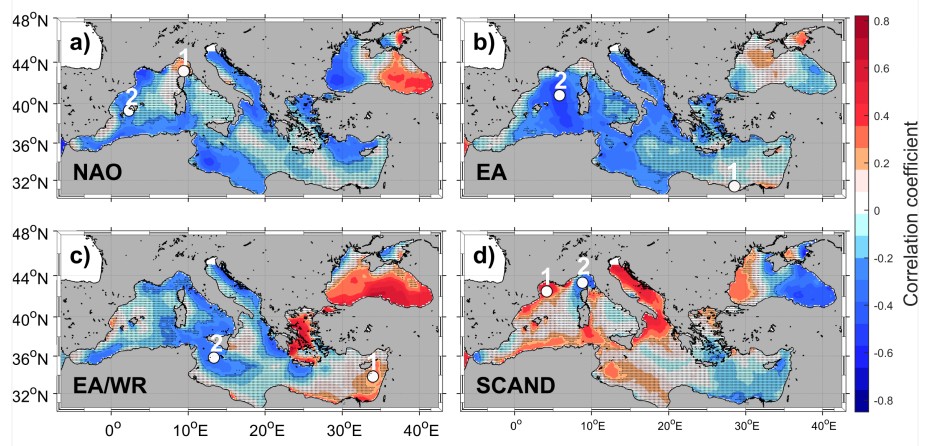

**Figure 8.** Pearson correlation coefficient of winter mean $99^{th}$ percentile SWH Mediterranean Sea series and a) NAO, b) EA, c) EA-WR and d) SCAND winter mean indices. No significant values at the $90\%$ confidence interval are dotted. The white points show 1) the maximum positive and 2) the maximum negative value of correlation coefficient.

## 3.2 Synoptic atmospheric composites associated to extreme wave patterns

The analysis of the atmospheric signature associated with extreme SWH is performed by computing the composites of extreme SWH, atmospheric mean sea level pressure (MSLP) and 10 m wind velocity (U10). The objective is to find the atmospheric pattern that is associated with the extreme winter waves. The procedure to build the composites is as follows:

- First, we select the locations with the highest correlations between $SWH_{99}$ and each of the atmospheric indices (points labeled as # 1 for maximum positive correlation and as #2 for maximum negative correlation in Fig. 7 and Fig. 8 for the
North Atlantic and Mediterranean Sea respectively).

- We select the time steps for which the original 3-hourly SWH time series at points #1 and #2 exceed $SWH_{99}$ (2 values each month are selected).

- Finally, we compute the composites for SWH, U10 and MSLP over the whole domain for all selected dates.

   Note that the locations numbered as #1 and #2 in each map represent the largest positive and negative correlations with
the corresponding index. The composite maps are thus interpreted as the synoptic patterns associated to positive and negative phases (respectively) of the climate index, leading to extreme waves.

**Atlantic ocean**

The composite for $U_{10}$ and MSLP built using location #1 (positive correlation between $SWH_{99}$ and NAO) (Fig. 9a) shows the typical configuration associated with the positive NAO phase that is characterized by low pressures across high latitudes





in the North Atlantic and high pressures over the central North Atlantic, the eastern United States and western Europe. This configuration leads a wind jet crossing the central North Atlantic whose fetch generates large waves at the western part of the British Islands and south of Iceland (SWH > 9 m). By contrast, in the south of the North Atlantic Ocean, in the Azores region, positive NAO phase results in low $SWH_{99}$ values (SWH ≈ 4 m). This pattern corresponds to the first EOF (Fig. 5a and Fig. 5-1 for the spatial mode and its amplitude respectively) when, for positive values of the PC, positive anomalies are

presented in the north part of the basin, and negative anomalies in the central part (the opposite for negative values of the PC). Composite for the positive phase of EA shows a similar structure than the one obtained for the positive phase of NAO, but with the North Atlantic cyclone shifted southwards and with the high pressures covering the entire Atlantic at 30° N (Fig. 9b). Maximum waves associated with the EA are obtained in the central Atlantic as the result of the southwards winds blowing from Greenland. This pattern corresponds to the second EOF (Fig. 5c and Fig. 5-2). Composite for the EA/WR positive phase shows

the low pressure system at 40° W, with the maximum extreme waves located to the east of Newfoundland (Fig. 9c). Finally, the atmospheric composite for the positive SCAND phase shows a cyclone (at 40° N) generating extreme waves smaller than the obtained with the previous three composites -values of SWH below 5m- Fig. 9d. This pattern is associated with the first EOF when its amplitude takes negative values (see Fig. 5b and Fig. 5-1) corresponding also with the atmospheric situation related with the negative phase of NAO (see Fig. 10a). For the negative EA phase, the cyclone is located between Greenland and

Iceland, generating a strong wind jet from the coast of Canada to Ireland (Fig. 10b). At this point, we want to remark that since here we are analyzing the negative correlations, the values displayed in Table 1 have to be changed in sign. The second EOF (Fig. 5c) according to Table 1 is related with the composite built for the maximum negative correlation between $SWH_{99}$ and the EA index (Fig. 10b). Composites for the negative phase of EA/WR, are characterized by a western shift of the Iceland low that generates strong zonal winds between 50°N and 60°N with maximum extreme waves located between south of Ireland and

North of Spain (Fig. 10c). The Iceland low for the negative SCAND phase, is shifted northeast of Iceland with winds blowing southwestwards and extreme waves located between Iceland and Great Britain (Fig. 10d). This composite is associated with the first EOF (Fig. 5b), thus having a correlation with the positive NAO phase (see also Fig. 9a for comparison).

**Mediterranean Sea**

In the Mediterranean Sea, for the positive correlations between indices and extreme waves, we choose locations near the coast

since negative correlations dominate the entire basin (Fig. 8). The atmospheric composite for the positive NAO phase displays a low pressure system in the north of Italy with associated eastwards winds in the western and central basins (Fig. 11a). These conditions are strongly associated with the atmospheric situations discussed in Trigo et al. (1999) for the cyclogenetic activity during winters. This composite is related with the distribution of extreme waves shown by the third EOF (Fig. 6d). Note that the amplitude of this mode is positively correlated with NAO according to Table 1. The composite for the positive EA phase

shows an intense cyclogenetic activity in the eastern Mediterranean Sea with its center of action over Cyprus which generates strong winds and waves north of Egypt (Fig. 11b). This index, as shown in Table 1, is negatively correlated with the amplitude of the second EOF whose pattern presents large values for the extreme wave anomalies over the eastern Mediterranean Sea (Fig. 6c); in other words, a positive EA results in larger $SWH_{99}$, in the eastern basin as displayed in Fig. 11b. Regarding

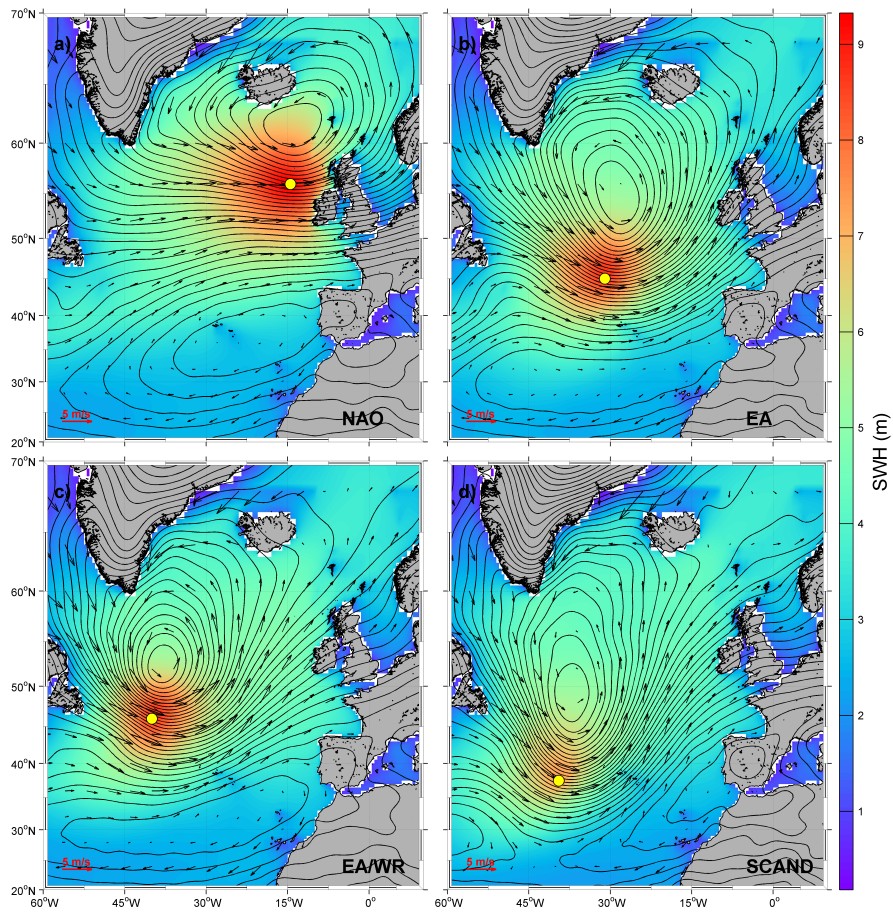

**Figure 9.** Winter atmospheric situations for the positive phase of a) NAO, b) EA, c) EA-WR and d) SCAND indices in the North Atlantic Ocean. The vectors represent the 10 m wind speed in m/s; the contours, the sea level pressure in Pa and the color range is the mean value of SWH in m. The red left bottom arrow represents the wind scale.

the positive EA/WR phase, the resulting composite presents a very similar pattern for surface pressure, winds and waves

than the one obtained for the positive EA phase (see Fig. 11b and Fig. 11c). The EA/WR index is also negatively correlated with the amplitude of the second EOF (Table 1) resulting in the same distribution of extreme waves as previously explained regarding the EA. Finally, the composite for the positive SCAND index displays a cyclonic structure on the northwestern part of the Mediterranean Sea -between Corsica and Sardinia- with winds blowing southwards at the Gulf of Lion (Fig. 11d). The SCAND index is positively correlated with the amplitude of the second EOF (Table 1), indicating larger/smaller SWH in the

western/eastern Mediterranean during its positive phase.

For the negative phases during winters, the composite for the NAO displays a weak cyclone over the Ligurian Sea (Fig. 12a). In this situation, however, the pressure gradient is weaker and due to the small fetch the resulting extreme waves are


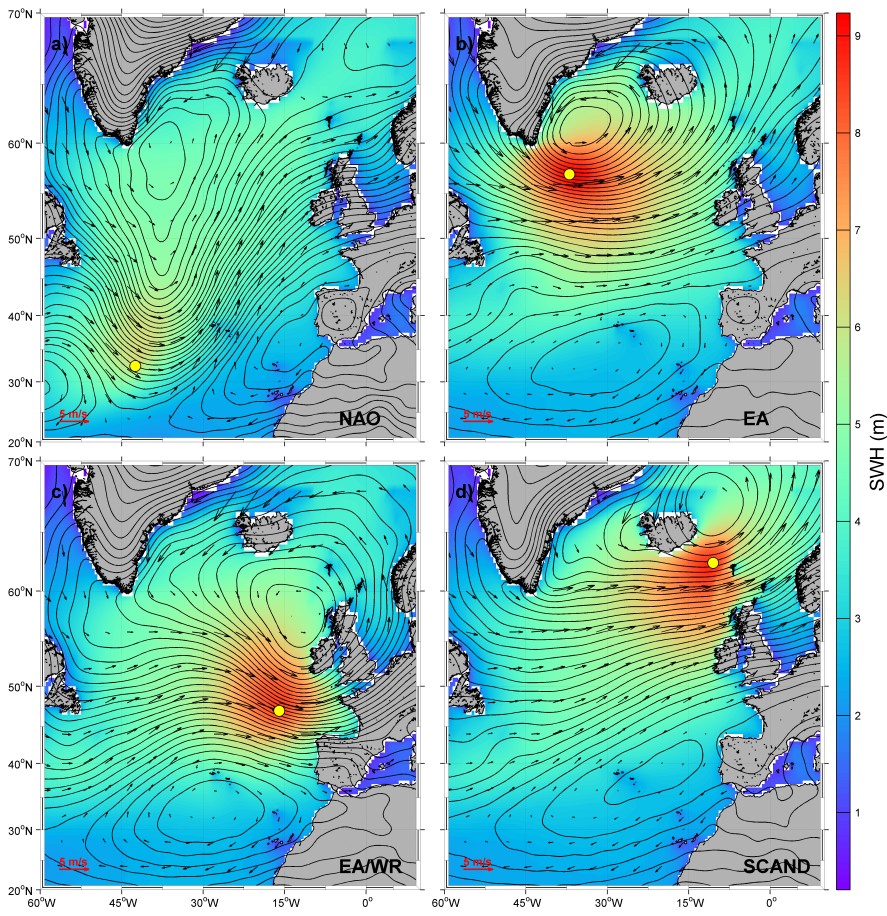

**Figure 10.** Winter atmospheric situations for the negative phase of a) NAO, b) EA, c) EA-WR and d) SCAND indices in the North Atlantic Ocean. The vectors represent the 10 m wind speed in m/s; the contours, the sea level pressure in Pa and the color range is the mean value of SWH in m. The red left bottom arrow represents the wind scale.

small (below 3m in SWH). Regarding the negative phase of the EA index, the composite shows a strong cyclone centered over Italy with a large pressure gradient over the northwestern Mediterranean Sea. This situation generates strong winds between
the Balearic Islands and Corsica and Sardinia, generating large waves in this area. EA index is negatively correlated with the amplitude of the second EOF (see Fig. 6c). For the negative phase of EA/WR, composite shows a low pressure system over the Ionian Sea with a strong pressure gradient between Sicily and Tunisia, resulting in large extreme waves in this passage (Fig. 12c). This index is negatively correlated with the amplitude of the second EOF (Table 1) suggesting that for positive anomalies of $SWH_{99}$ (see Fig. 6c) a negative phase of the EA/WR index results in an increase of extreme waves. Finally, the composite
for the negative phase of the SCAND index displays a low pressure system over north of Italy. Although the pressure gradient associated with this low system is intense north of Corsica, the small fetch impels the formation of extreme waves (Fig. 12d).





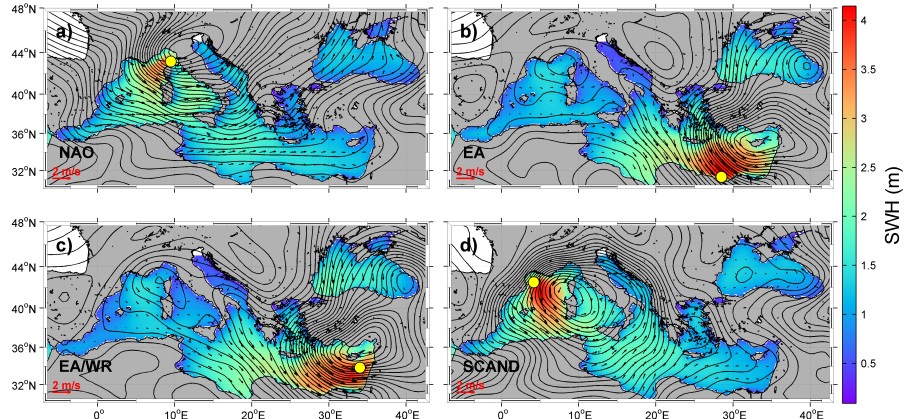

**Figure 11.** Winter atmospheric situations for the positive phase of a) NAO, b) EA, c) EA-WR and d) SCAND indices in the Mediterranean Sea. The vectors represent the 10 m wind speed in m/s; the contours, the sea level pressure in Pa and the color range is the mean value of SWH in m. The red left bottom arrow represents the wind scale.

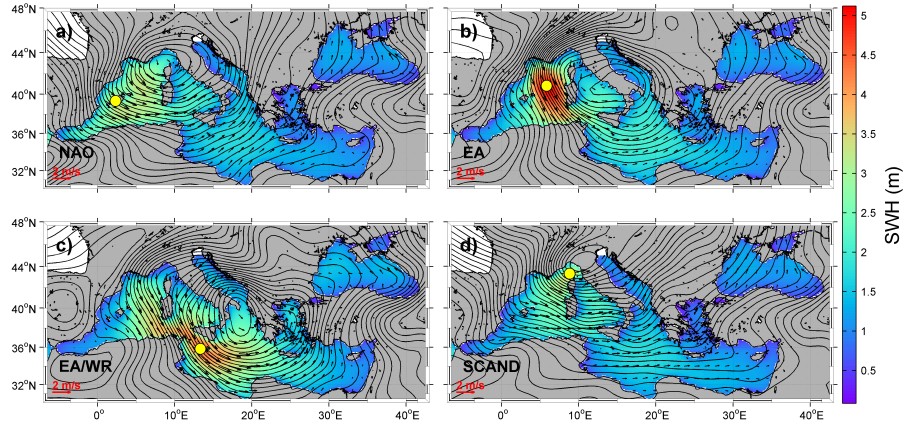

**Figure 12.** Winter atmospheric situations for the negative phase of a) NAO, b) EA, c) EA-WR and d) SCAND indices in the Mediterranean Sea. The vectors represent the 10 m wind speed in m/s; the contours, the sea level pressure in Pa and the color range is the mean value of SWH in m. The red left bottom arrow represents the wind scale.

## 4 Summary and conclusions

This work presents a new methodology to study the extreme wave climate and the atmospheric synoptic conditions responsible for the extreme waves. Winds and pressure are obtained by computing the composites corresponding to the monthly values
of the $99^{th}$ percentile of the significant wave height. As a result, it is possible to infer changes in the location and intensity of extreme waves through the understanding of the variability of climatic patterns (widely studied). This approach could be





of interest for all the activities related to the prognostic of extreme waves, such as, the design of offshore structures, among others.

The study is focused on the North Atlantic Ocean and the Mediterranean Sea, although the methodology can be extrapolated
to any region in order to get a deeper insight on the seasonal and interannual variability of extreme wave climate. In the present work, the interannual variability has been analyzed using Empirical Orthogonal Functions, which have been correlated against the four main climate indices of variability in the area, i.e., NAO, EA, EA/WR and SCAND. Finally, the most reliable atmospheric situation associated with each climatic pattern is discussed using the described methodology.

The extreme waves climate has a large seasonal signal both in the North Atlantic Ocean and in the Mediterranean Sea. Our
results also indicate a large intra-annual variability in the central part of the Mediterranean Sea and lower in the Alboran and in the Ligurian sub-basins, in agreement with Sartini et al. (2017) where they exhibited different degrees of seasonality depending on the main mesoscale meteorological features of the locations analyzed. Concerning long term trend of extreme waves, it is predominantly negative although there are some areas, as in the center of the North Atlantic Ocean or in the Aegean Sea, where the value of the tendency is positive. These results are not in line with the studies of Young et al. (2011) and Young and Ribal
(2019) because here, we assess only the extreme waves values during the winter season, when most of the maximum SWH occur.

Regarding climatic modes of variability, we found that the NAO and the SCAND indices are the leading modes of climatic variability affecting extreme waves in the North Atlantic Ocean during winters. The positive NAO phase increases extreme waves in the northern North Atlantic while the negative NAO phase results in an increase of extreme waves in the southern
North Atlantic in accordance with Hurrell et al. (2003). By contrast, a positive SCAND index increases extreme waves in the southern North Atlantic while a negative SCAND index increases the extreme waves in the north part of the North Atlantic Ocean, as Martínez-Asensio et al. (2016) also pointed out. To a lesser extent, the EA also influences the extreme waves in the North Atlantic Ocean, as Izaguirre et al. (2010) concluded, too. While the positive EA phase drives extreme wave climate in the central North Atlantic, the negative phase controls extreme wave climate at higher and lower latitudes (see Fig. 7b).
The interannual variability of extreme waves during winters in the Mediterranean Sea is dominated, to a large extent, by the negative phase of EA with larger effect in the western basin. Positive NAO phase has also influence on extreme waves although they are smaller in the whole Mediterranean Sea.

Caires et al. (2006) reported that the wave climate is expected to change by a small amount in response to climate change (below $5\%$ between 1990 and 2080). The results presented here could be used to project climate, and to develop appropriate
studies for coastal protection improving numerical models and to define long term wave energy conversion strategies; since the climatic patterns of NAO will dominate in a larger extent the extreme waves climate at the North Atlantic Ocean in future scenarios, according to Gleeson et al. (2017).

*Data availability.* All data are accessible from https://polar.ncep.noaa.gov/waves/hindcasts/nopp-phase2.php and from https://www.cpc.ncep.noaa.gov/data/teledoc/telecontents.shtml



*Author contributions.* A.O. conceived the idea of the study with the support of V.M., G.S. and M.M.; A.O. and V.M. developed the methodology with the support of M.M. and G.S.; V.M. produced the results with the support of A.O. and M.M.; M.M. and G.S. analyzed the results with the support of A.O and V.M. All authors contributed to write the MS.

*Competing interests.* AO is a member of the editorial board of Ocean Dynamics and Frontiers in Marine Sciences. MM is a member of the editorial board of Frontiers in Marine Sciences

*Acknowledgements.* Authors acknowledge financial support from MINECO/FEDER through project MORFINTRA/MUSA (CTM2015-66225-C2-2-P) and from Ministerio de Ciencia, Innovación y Universidades through MOCCA project (RTI2018-095441-B-C21). V. Morales-Márquez is supported by an FPI grant from the Ministerio de Ciencia, Innovación y Universidades. In addition, V. Morales-Márquez acknowledges the 2019 EGU-OSPP winning award for this work. This work was partially performed while A. Orfila was a visiting scientist at the Earth, Environmental and Planetary Sciences Department at Brown University through a Ministerio de Ciencia, Innovación y Universidades fellowship (PRX18/00218).



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
