# Peer review of "Extreme waves and climatic patterns of variability in the Eastern North Atlantic and Mediterranean basins"

_Ocean Science, 2020_

## Referee Comment (RC1) · Anonymous Referee #1 · 29 Jun 2020

The manuscript "Extreme waves and climatic patterns of variability in the Eastern North Atlantic and Mediterranean basins" by Morales-Márquez et al. describes an analysis of the 99percentile of significant wave height focused during winter months over NE Atlantic and Mediterranean Sea. The study is interesting and some outcomes show novel information and wave climate variability patterns (eg. the contribution of seasonality into variability, historical trends over the 30-year period used, correlation with climate indices). From my point of view, however, the manuscript requires an improvement in the wording to better describe some technical steps. One of the main weaknesses of the work is that it seems that the SWH has not been validated against observations, neither the SHW99 magnitude nor its variations.

[Figure]

Specific comments: i. English grammar needs to be checked

ii. Lines 11-12 "Besides, extreme waves influence the upper ocean by enhancing vertical mixing through the Stokes layer". A reference of the statement would be desirable.

iii. Line 13. I do not completely understand the message about the role of the extreme waves in coastal flooding at 'intra-annual' scale. Please, consider rewriting.

iv. References of the journal articles are odd worded at the end of the manuscript, making difficult find the referenced information sources.

v. I suggest removing 'methods' of the title of section 2 since EOFs, correlation significance estimation, composites, etc. methods are not described in this section

vi. Section 2.1 "Waves and Atmospheric Data" requires organisation and adding relevant details for the analysis of extreme wave climate. I am confused by some aspects that are ambiguously mentioned. Some examples: a) What is the time resolution of the used winds to generate the waves by forcing WaveWatch model? £winds fields at 0.5deg are used over the Mediterranean Sea? £is this spatial resolution enough to simulate wave extremes? b) What is the time resolution of SWH from the database? Averaged 3hourly values? One hourly value each 3 hours? c) £has the used wave data source been validated in the study area? A comparison against buoy records in the analyzed domains shown in figure 1 is crucial to validate the further analysis of extreme waves

vii. I wonder how robust figure 2c and 2d are, as they are calculated from the maximum 99percentile value of a month and year. Are the monthly spatial patterns preserved for the averaged month of highest 99percentile value?

viii. Is semi-annual cycle statistically significant in the regression model? (eq.1). Panels of figure 3 do not show clear semiannual cycles for points 1 & 2. Maybe the variance reduction is only due to annual cycle

ix. Lines 90-93. Please, review the grammar. I do not completely understand the

insights in the sentences. How 'generation wave areas' have been estimated/detected? What does development wave area mean?

x. I do not see the relationship (alignment) of the historical estimated SWH99 trends (1979-2009) with future projections of surface wind (referenced from Gallagher et al. 2016). Please, clarify this statement

xi. Line 116. Please, provide details about how the 5yr-periodicity is estimated. I cannot see clearly in Fig.5-1

xii. White color of colorbar in figures 7 & 9 must be centered on zero

xiii. Line 186-87. Please, clarify the step 2 to build the composite. I do not understand what the authors refer as 'time steps' and why only 2 values per month are obtained

xiv. Lines 189-91. As far as I understand how the composite is built, the composite maps only represent spatial patterns for a target location with a high correlation between winter SHW99 and a climate index. They are not synoptic maps (a map for a given moment of time)

xv. Figures 9 & 10. The magnitude of the wind vector and SLP would help the reader to better understand the resulting maps

---

## Author Comment (AC1) · 23 Jul 2020

The reply is inside the supplementary documents.

Please also note the supplement to this comment: https://os.copernicus.org/preprints/os-2020-34/os-2020-34-AC1-supplement.pdf

[Figure]

**Fig. 1.** Location of study zones. a) Eastern North Atlantic Ocean. b) Mediterranean Sea.

[Figure]

**Fig. 2.** Trend of the monthly 99th percentile SWH during winters (DJFM) in cm/year. No significant values at the 90% confidence interval are dotted.

[Figure]

**Fig. 3.** Pearson correlation coefficient of winter mean 99th percentile SWH North Atlantic series and a) NAO, b) EA, c) EA-WR and d) SCAND winter mean indices. No significant values are dotted.

[Figure]

**Fig. 4.** Pearson correlation coefficient of winter mean 99th percentile SWH Mediterranean Sea series and a) NAO, b) EA, c) EA-WR and d) SCAND winter mean indices. No significant values are dotted.

**Supplement:**

**Referee #1**

Q1. The manuscript "Extreme waves and climatic patterns of variability in the Eastern North Atlantic and Mediterranean basins" by Morales-Márquez et al. describes an analysis of the 99percentile of significant wave height focused during winter months over NE Atlantic and Mediterranean Sea. The study is interesting and some outcomes show novel information and wave climate variability patterns (eg. the contribution of seasonality into variability, historical trends over the 30-year period used, correlation with climate indices).

Reply #1. We deeply thank Referee's comments and the effort that she/he made in reviewing carefully our work. We have introduced in the new version of the Manuscript (hereinafter Ms) all points raised in the review. We sincerely think that the new version of the Ms has been improved, thanks to this discussion.

Q2. From my point of view, however, the manuscript requires an improvement in the wording to better describe some technical steps. One of the main weaknesses of the work is that it seems that the SWH has not been validated against observations, neither the SHW99 magnitude nor its variations.

Reply #2. As suggested, all technical aspect raised by the Reviewer have been modified following her/his suggestions. In particular we have included a comparison between the wave reanalysis and available in situ buoys. All Referee's concerns are explained in detail in the following replies.

**SPECIFIC COMMENTS**

Q3. English grammar needs to be checked.

*Reply* #3. *The Ms. has been revised in detail and typos corrected.*

Q4. Lines 11-12 "Besides, extreme waves influence the upper ocean by enhancing vertical mixing through the Stokes layer". A reference of the statement would be desirable.

*Reply* #4. As suggested by the Reviewer, a new reference has been included in this sentence of the manuscript. Lines #11-12 read,

"Besides, extreme waves influence the upper ocean by enhancing vertical mixing through the Stokes layer (Polton et al., 2005).

Q5. Line 13. I do not completely understand the message about the role of the extreme waves in coastal flooding at 'intra-annual' scale. Please, consider rewriting.

Reply #5. Coastal flooding depends on the combined effect of Sea Level, Storm Surges and Wave Setup. While the first impact is a large-scale effect, surges and wave setup have a clear seasonality (see Figure 2 panels c and d). Thus, larger impacts in flooding are related with maximum waves and large surges that are intra-annual processes.

Q6. References of the journal articles are odd worded at the end of the manuscript, making difficult find the referenced information sources.

*Reply* #6. *We followed the Ocean Science template for the References.*

Q7. I suggest removing 'methods' of the title of section 2 since EOFs, correlation significance estimation, composites, etc. methods are not described in this section.

Reply #7. We modified the Section title as "Data and extreme wave values".

Q8. Section 2.1 "Waves and Atmospheric Data" requires organisation and adding relevant details for the analysis of extreme wave climate. I am confused by some aspects that are ambiguously mentioned. Some

examples: a) What is the time resolution of the used winds to generate the waves by forcing WaveWatch model? £winds fields at 0.5deg are used over the Mediterranean Sea? £is this spatial resolution enough to simulate wave extremes? b) What is the time resolution of SWH from the database? Averaged 3hourly values? One hourly value each 3 hours? c) £has the used wave data source been validated in the study area? A comparison against buoy records in the analyzed domains shown in figure 1 is crucial to validate the further analysis of extreme waves.

Reply #8.

*a)* The temporal resolution of the winds forcing the wave model is 1 hour. It is defined at lines #54-57 of the Ms.:

"This dataset (i.e. WAVEWATCH III 30-yearHindcast Phase 2, (Chawla et al., 2012)) has been generated by forcing the "state-of-the-art" wave model WAVEWATCH III (Tolman, 2009) with 10-m height high-resolution wind fields from the NCEP Climate Forecast System Reanalysis and Reforecast (CFSRR) a 30-year homogeneous data set of hourly 1/2° spatial resolution winds."

Regarding the wind resolution at the Mediterranean Sea the hindcast fits better in this basin than in the North Atlantic Ocean according to the buoys data from CMEMS (see Table 1 of the Ms), having largely been used in the North Atlantic to analyze the mean and the extreme waves (see for instance Forte, M.F. et al., 2012; and Gonçalves, M. et al., 2018).

b) The wave model provides outputs every 3 hours, which is largely accepted to be a sea state.

c)Thanks to referee's suggestions we compare the hindcast at the Mediterranean Sea and NE Atlantic Ocean with buoys available at Copernicus (https://marine.copernicus.eu/). In addition, we write the following sentences in order to include this comparison in the Ms. (Lines #71-79):

"The hindcast is validated at the two basins with available buoys following the methodology described in Morales Márquez et al. (2018). For 13 buoys in the Atlantic and 11 in the Mediterranean Sea, we compute the correlation coefficient ( $R^2$ ), the Scatter Index (SCI and the Relative Bias (RB) defined as:

$$R^2 = \frac{\text{Cov(b,m)}}{\sigma_{\rm b}\sigma_{\rm m}},$$

$$SCI = \frac{\mathrm{rms}_{\mathrm{m-b}}}{\mathrm{max}(\mathrm{rms}_{\mathrm{b}}, |\langle \mathrm{b} \rangle|)},$$

$$RB = \frac{\langle \mathbf{m} - \mathbf{b} \rangle}{\max(\mathrm{rms}_{\mathbf{b}}, |\langle \mathbf{b} \rangle|)}$$

where m is the hindcast at buoy location and b the buoy data.

Table 1 shows the comparison between the WAVEWATCH III 30 hindcast and the buoys fromCopernicusMarineenvironmentmonitoringservice(CMEMS,https://marine.copernicus.eu/).See Fig. 1 for buoy locations.

Table 1. Statistical comparison between the WAVEWATCH III 30 hindcast and CMEMS buoys.

| Basin             | $R^2(\%)$        | SCI           | Relative bias    |
|-------------------|------------------|---------------|------------------|
| Mediterranean Sea | $73.09\pm0.17$   | $0.39\pm0.00$ | $-0.13 \pm 0.00$ |
| North Atlantic    | $72.67 \pm 0.96$ | $0.33\pm0.02$ | $0.16\pm0.02$    |

Additionally, Figure 1 has been modified in order to show location of buoys.

Q9. I wonder how robust figure 2c and 2d are, as they are calculated from the maximum 99percentile value of a month and year. Are the monthly spatial patterns preserved for the averaged month of highest 99percentile value?

Reply #9. Figure 2 shows the maximum value of all monthly 99 percentiles. Panels a and b correspond to the value and panels c and d to the month when it occurs. The monthly patterns are not preserved (each grid point has its maximum at a specific month). The interest of this Figure is thus the homogeneity in the obtained months since the most of extreme waves are concentrated in few months (Dec-Mar).

Q10. Is semi-annual cycle statistically significant in the regression model? (eq.1). Panels of figure 3 do not show clear semiannual cycles for points 1 & 2. Maybe the variance reduction is only due to annual cycle

Reply #10. We compute the level of significance for the annual and semiannual cycle being both significant in the whole basins. As Referee points out, most of the variance reduction is due to the annual cycle but the semiannual variation is also significant both in the North Atlantic Ocean and in the Mediterraean Sea. In addition, we show that the intra-annual frequencies of the SWH99 signal have to be removed if we want to analyze the long-term variability of the extreme waves.

Q11. Lines 90-93. Please, review the grammar. I do not completely understand the insights in the sentences. How 'generation wave areas' have been estimated/detected? What does development wave area mean?

Reply #11. Thank you for your advice. We rewrite the lines #99-102 of the Ms. as,

"In the Mediterranean Sea, there are two different areas in terms of seasonality. One is located in the central basin where seasonality explains up to a 70% of extreme waves and the other located in the Gulf of Genova and the Alboran Sea, where seasonality explains less than 10% of the signal (Fig. 3b). These areas are very active in terms of ciclogenetic activity (Trigo et al., 2002) and thus the seasonal signal is relatively less important here.

Regarding the estimation of the generation areas, we distinguish between Sea and Swell waves the first generated by local winds and the second including intensity, duration and distance (waves travelling across large areas). Swell waves are more ordered, presenting a more regular appearance with larger periods and wavelengths in a narrower frequency range than Sea type waves. The wave generation areas can be estimated analyzing wave features (SWH and wavelength, or period since they are related through the dispersion relationship) and local winds.

In these lines of the Ms., we explain how the local wind dominates the areas where the seasonality describes a smaller value of the extreme wave; corresponding with the generated waves areas. On the other hand, the higher values of the seasonality take place in regions where the waves can be developed better because they travel large distances. In these areas the dominant sea state corresponds to Swell.

Q12. I do not see the relationship (alignment) of the historical estimated SWH99 trends (1979-2009) with future projections of surface wind (referenced from Gallagher et al. 2016). Please, clarify this statement.

Reply #12. Figure 3a and b of Gallagher et al, 2016., show the projected changes for the period of 2070-2099 relative to 1980-2009. There, they obtain a similar wind variation pattern than the trend of  $SWH_{99}$  map that we present. We want to remark that the tendency of the  $SWH_{99}$  most likely will follow the same pattern that we have calculated.

We have rewritten these lines in order to clarify it, in #108-110 of the Ms. as,

"This aligns with the obtained results in Gallagher et al. 2016., where the future projections of mean surface wind show an average decrease over the North Atlantic Ocean for winter season, so likely the extreme waves continue with the same pattern in terms of long-term variability.".

Q13. Line 116. Please, provide details about how the 5yr-periodicity is estimated. I cannot see clearly in Fig.5-1.

*Reply* #13. We modified the Ms and we now explain that we computed it through a FFT analysis of the PC. For that, we add this information in #125-126 lines of Ms. as,

"The first EOF, which explains a 28.5% of the winter SWH99, presents a periodicity in its PC around 5 years (calculated through FFT analysis of Fig. 5-1)."

Q14. White color of colorbar in figures 7 & 9 must be centered on zero.

Reply #14. This has been fixed. In addition, we change the colors of the colorbar in order to distinguish better the values of the correlation. We modify also the colorbar of the Fig. 4 in order to have the same colors tones in all the figures.

Q15. Line 186-87. Please, clarify the step 2 to build the composite. I do not understand what the authors refer as 'time steps' and why only 2 values per month are obtained.

*Reply* #15. *We clarify the methodology used for obtaining composites with the following example for the positive NAO phase. The steps are the following:*

1. We select the grid point with maximum correlation (both positive and negative values) between SWH99 and the climatic index.

In this case point #1 presents the maximum positive correlation with the NAO and for simplicity we plot SWH for January 1979 (the process is done for the whole time series, repeating this methodology for all the months).

2. From the 3-hourly dataset the following data are selected:

**$SWH_{3-hours} \ge SWH_{99 month}$**

As the SWH time series are of 3-hourly time step, we have 224 data in months of 28 days, 232 data in months with 29 days, 240 data in months with 30 days, and 248 in months with 31 days. The 99th percentile is defined as the value that only the 1% of the time series exceed it. And with our time step, the 1% of the number of data for each month those are higher than SWH99 is only 2 values.

For January 1979 and for the positive phase NAO composite is concerned, the values that exceeded than  $SWH_{99}$  for that month occur at 12h and at 15h on  $28^{th}$  of January 1979.

---

## Referee Comment (RC2) · David Woolf (Referee) · 28 Jul 2020

The study investigates the relationship of extreme wave heights to atmospheric modes based on a high-quality wave hindcast. The study is described concisely and clearly. There are not any great revelations, but a useful study is reported fairly and competently in an appropriate form and to a suitable journal. I have some detailed comments, which are given below, but I am content for a revised version to be published after reasonable attention to all reviewer comments.

I do not have a problem with calculating the linear trends by a simple method and reporting these (Figure 4 and associated text), But I'd urge caution in interpretation.

[Figure]

Firstly, there is likely to be some autocorrelation in the atmospheric forcing (and thus wave heights), which makes the independence of values assumed in simple regression doubtful. Secondly, the particular time period, 1979-2009 is pertinent; different periods would show different patterns. A similar weakness in estimation of statistical significance is apparent in the use of "t-values" for the relationship of PCs to climate indices (lines 130-136), but again that is a minor objection and should not discourage publication. It is possible to take the statistical analysis further, for example through a wavelet analysis of the wave height - climate index relationship (I have seen this for sea level, but I am not aware of such an analysis for wave heights), but that is for another paper.

My general impression is that, as for some previous studies, the relationship of extreme waves in the North Atlantic to NAO is very convincing, but (albeit with calculated significance) the other relationships are rather weak (possibly with the exception of SCAND). It seems to me to be open to debate if we understand the extreme wave heights better from modest correlations to atmospheric indices. The relationship to composite was more interesting and in some respects was more convincing. For example, the relationship to EA is physically sensible and quite satisfying in this form. The sections from lines 193-217 and from 219-246 are rather monotonous and not very effective in communication. I suggest finding a more engaging method of communicating this information, perhaps a Table?

I do not have any strong objections to the content of "Summary and conclusions" though I can be counted as a sceptic regarding simple projections of NAO behaviour and their utility in projections of extreme waves.

The abstract adequately describes the topic and principal results, but gives no explanation of the methodology beyond "31-year wave model hindcast". I suggest adding another sentence. Line 64-65. "we assume that wave climate is constant for 3 hours". I interpret that phrase as an assumption of an autocorrelation period of 3 hours, is that correct? Was the data analysed to reach this conclusion? Does it have any implications beyond simply informing using 3-hour data? Line 76 and following: There are ~240 3-hour values in each calendar month. How exactly is the 99th percentile calculated? (An interpolation between the second and third highest values for each month?) Line 82. How good is a fit of annual and semiannual sinusoidal to the seasonality? Was there any analysis for higher harmonics? Line 93. spelling "cyclogenetic" Line 95. Perhaps change "adjusted though a first order polynomial . . ." to "fitted by a linear regression in time"? Line 101. Change "northern" to "north" Line 116. "periodicity . . . around 5 years" Not wrong, but perhaps risky? I would generally avoid talking about periodicity unless there is a very strong case. Line 145, ". . . being the rest of correlations marginal". I could not make sense of this line! Line 196 ". . . leads a wind jet". I suggest "this composite is characterise by a strong westerly wind stream . . ." N.B. A similar relationship was demonstrated dynamically by Wolf and Woolf (2006; GRL 33(6)). Add gratitude to NCEP and NOAA CPC for data in Acknowledgments?

---

## Author Comment (AC2) · 20 Aug 2020

**Referee #2: David Woolf**

Q1. The study investigates the relationship of extreme wave heights to atmospheric modes based on a high-quality wave hindcast. The study is described concisely and clearly. There are not any great revelations, but a useful study is reported fairly and competently in an appropriate form and to a suitable journal. I have some detailed comments, which are given below, but I am content for a revised version to be published after reasonable attention to all reviewer comments.

> *Reply #1. We sincerely thank reviewer for his constructive comments and the effort that he made in reviewing carefully our work. We deeply think that thanks to this revision the new version of the manuscript (hereinafter Ms.) improved significantly.*

**SPECIFIC COMMENTS**

Q2. I do not have a problem with calculating the linear trends by a simple method and reporting these (Figure 4 and associated text), But I'd urge caution in interpretation. Firstly, there is likely to be some autocorrelation in the atmospheric forcing (and thus wave heights), which makes the independence of values assumed in simple regression doubtful. Secondly, the particular time period, 1979-2009 is pertinent; different periods would show different patterns.

> *Reply #2. We totally agree with the referee's comment. We consider relevant to write a new sentence in the Ms. in order to clarify this issue (lines #116-117):*
>
> *"These calculations are restricted to the time period corresponding to the atmospheric forcing of the WAVEWATCH III and different patterns could be obtained for different periods".*

Q3. A similar weakness in estimation of statistical significance is apparent in the use of "t-values" for the relationship of PCs to climate indices (lines 130-136), but again that is a minor objection and should not discourage publication. It is possible to take the statistical analysis further, for example through a wavelet analysis of the wave height - climate index relationship (I have seen this for sea level, but I am not aware of such an analysis for wave heights), but that is for another paper.

> *Reply #3. We agree with the referee's comment again. The lines #151-152 of the Ms. have been rewritten in order to note this issue, as:*
>
> *"These significance values are particular for this study since they depend on the data used and the analyzed time period."*
>
> *We agree that a wavelet analysis would complement these kind of studies. In order to keep the focus of the work we prefer to perform them in a new Ms. However, we pointed this comment in the Summary and conclusions section. Lines #290-292 of Ms. read:*
>
> *"For future studies, a wavelet coherence analysis (Torrence and Compo, 1998) between the main climatic indices and the extreme waves will provide additional information on the dominant modes of variability and how they vary in time."*

Q4. My general impression is that, as for some previous studies, the relationship of extreme waves in the North Atlantic to NAO is very convincing, but (albeit with calculated significance) the other relationships are rather weak (possibly with the exception of SCAND).

> *Reply #4. Yes, the NAO index is the most relevant climatic mode of the North Atlantic Ocean, being noteworthy the effect of the SCAND index. However, we include the rest of climatic indices (EA and EA/WR) analysis because they have a significant influence on the extreme waves in the Mediterranean basin. And, as we mention in the Ms., we use the North Atlantic Ocean study as a "validation" of the methodology developed and then applying it in the Mediterranean Sea, since the North Atlantic wave climate has been largely analyzed in previews studies.*

Q5. It seems to me to be open to debate if we understand the extreme wave heights better from modest correlations to atmospheric indices. The relationship to composite was more interesting and in some respects was more convincing. For example, the relationship to EA is physically sensible and quite satisfying in this form.

> *Reply #5. We agree with the referee in this aspect, we think that the composite analysis is the most innovative and interesting study in this paper, being a more suitable way to relation the climatic modes with the extreme waves. However, the proposed methodology needs to compute the correlation in order to select the spatial point where the climatic patterns have a higher influence. In addition, it can be useful for the better understanding of the relation of the climatic indices and the extreme waves for the reader.*

Q6. The sections from lines 193-217 and from 219-246 are rather monotonous and not very effective in communication. I suggest finding a more engaging method of communicating this information, perhaps a Table?

> *Reply #6. We are not sure if the referee talks about the description of the composites or about the relation between the composites and the EOFs. If it is the first case, we consider these sections as descriptive (we only describe the composite results; Fig. 9-12), being the composite figures the clearer and concise way to transmit the information. If, on the other hand, he refers to the relation with the EOFs and PCs, the table which show that is the Table 2 (in new Ms., old Table 1) where the correlation between the main climatic indices and the principal modes appear. We want to remark that we obtain the same results if we compare the signals of PCs with the different climatic indices and the EOFs maps with the composites.*

Q7. I do not have any strong objections to the content of "Summary and conclusions" though I can be counted as a sceptic regarding simple projections of NAO behaviour and their utility in projections of extreme waves.

> *Reply #7. We think that the climatic indices signal is more stable than the extreme waves, and it can be a suitable way to know any probable behavior of them. However, we are aware that when we mention the future projections, we are talking about statistics, but if we improve somehow the extreme waves prognostic, this paper will have met its target.*

Q8. The abstract adequately describes the topic and principal results, but gives no explanation of the methodology beyond "31-year wave model hindcast". I suggest adding another sentence.

> *Reply #8. The following sentence has been added in lines # 7-10 of the new Ms.,*
>
> > *"A new methodology for analyzing the atmospheric signature associated with extreme waves is proposed. The method obtains the composites of Significant Wave Height (SWH), mean sea level pressure (MSLP) and 10 m-height wind velocity (U10) using the instant when specific climatic indices have the stronger correlation with extreme waves."*

Q9. Line 64-65. "we assume that wave climate is constant for 3 hours". I interpret that phrase as an assumption of an autocorrelation period of 3 hours, is that correct? Was the data analysed to reach this conclusion? Does it have any implications beyond simply informing using 3-hour data?

> *Reply #9. For a limited period of time and in a particular geographical region, wave conditions vary in a stationary way. For this reason, it is commonly assumed that a sea state remains stationary for 3-6 hours. The objective of the sentence in the Ms. is only to explain that the time step is suitable in order to do the presented analysis in this paper, since we follow the common physical assumption of the wave state where the waves features are stationary during 3 hours (the time step of our dataset).*

Q10. Line 76 and following: There are ~240 3-hour values in each calendar month. How exactly is the 99th percentile calculated? (An interpolation between the second and third highest values for each month?)

*Reply #10. The 99$^{th}$ percentile is calculated as the value that only the 1% of the monthly data exceed it. Therefore, this value is interpolated between the second and third highest values for each month since we have 1 data each 3 hours (with ~240 data/month only 2.4 data exceed the 99$^{th}$ percentile value).*

Q11. Line 82. How good is a fit of annual and semiannual sinusoidal to the seasonality? Was there any analysis for higher harmonics?

*Reply #11. Fitting the seasonality of the monthly SWH$_{99}$ series as a cosine function through a least squares adjustment, we remove the annual and semiannual frequencies in the both basins. We check this method with a fast Fourier transform analysis, comparing both signal (with and without seasonality) obtaining that the annual and semiannual periods are removed. In addition, we verify that there is any dominant higher frequency in time series. This is because when computing the extreme waves any periodicity inside a month is erased.*

Q12. Line 93. spelling "cyclogenetic"

*Reply #12. This has fixed.*

Q13. Line 95. Perhaps change "adjusted though a first order polynomial : : :" to "fitted by a linear regression in time"?

*Reply #13. This has been fixed.*

Q14. Line 101. Change "northern" to "north".

*Reply #14. This has been fixed.*

Q15. Line 116. "periodicity : : : around 5 years" Not wrong, but perhaps risky? I would generally avoid talking about periodicity unless there is a very strong case.

*Reply #15. We modified the Ms and we now explain that we computed it through a FFT analysis of the PC. For that, we add this information in #129-130 lines of Ms. as,*

*"The first EOF, which explains a 28.5% of the winter SWH99, presents a periodicity in its PC around 5 years (calculated through FFT analysis of Fig. 5-1)."*

Q16. Line 145, ": : : being the rest of correlations marginal". I could not make sense of this line!

*Reply #16. We want to remark that the rest of the correlations are not significant. However, we agree with the referee's comment in that this line is misleading, and thus we remove this part of the sentence.*

Q17. Line 196 ": : : leads a wind jet". I suggest "this composite is characterized by a strong westerly wind stream : : :" N.B. A similar relationship was demonstrated dynamically by Wolf and Woolf (2006; GRL 33(6)).

*Reply #17. This has been changed.*

Q18. Add gratitude to NCEP and NOAA CPC for data in Acknowledgments?

*Reply #18. Thank for the referee's advice. We write the following sentence in lines #309-310 of Ms:*

*"In addition, authors thank NCEP and NOAA CPC for the free available data that have been used in this article."*

---

## Author Response (AR2)

**Topic Editor**

Q1 The authors have addressed the revisions of the reviewers satisfactorily. There are still a few spelling mistakes and slightly odd phrasing but I hope this will be corrected at the next stage, e.g. spelling of 'deeply' in line 300.

*Reply #1. Dear Editor,*

*We thank the response regarding manuscript. os-2020-34. We have undertaken a Review of the manuscript and corrected all typos. We hope that with this action the Ms. will be finally ready for publication.*

*With best regards,*

*The authors*